# The estimated burden of scrub typhus in Thailand from national surveillance data (2003-2018)

Tri Wangrangsimakul[1,2]*, Ivo Elliott[2,3], Supalert Nedsuwan[4], Rawadee Kumlert[5], Soawapak Hinjoy[6], Kittipong Chaisiri[7], Nicholas P. J. Day[1,2], Serge Morand[8]

1 Mahidol-Oxford Tropical Medicine Research Unit (MORU), Faculty of Tropical Medicine, Mahidol University, Bangkok, Thailand, 2 Centre for Tropical Medicine and Global Health, Nuffield Department of Clinical Medicine, University of Oxford, Oxford, United Kingdom, 3 Lao-Oxford-Mahosot Hospital-Wellcome Trust Research Unit, Microbiology Laboratory, Mahosot Hospital, Vientiane Capital, Lao People's Democratic Republic, 4 Social and Preventative Medicine Department, Chiangrai Prachanukroh Hospital, Ministry of Public Health, Chiangrai, Thailand, 5 The Office of Disease Prevention and Control 12 Songkhla Province, Department of Disease Control, Ministry of Public Health, Nonthaburi, Thailand, 6 Department of Disease Control, Ministry of Public Health, Nonthaburi, Thailand, 7 Department of Helminthology, Faculty of Tropical Medicine, Mahidol University, Bangkok, Thailand, 8 CNRS ISEM-CIRAD ASTRE, Faculty of Veterinary Technology, Kasetsart University, Bangkok, Thailand

* tri@tropmedres.ac

**Data Availability Statement:** All relevant data are within the manuscript and its Supporting Information files.

## Abstract

### Background

Scrub typhus is a major cause of acute febrile illness in the tropics and is endemic over large areas of the Asia Pacific region. The national and global burden of scrub typhus remains unclear due to limited data and difficulties surrounding diagnosis.

### Methodology/Principal findings

Scrub typhus reporting data from 2003–2018 were collected from the Thai national disease surveillance system. Additional information including the district, sub-district and village of residence, population, geographical, meteorological and satellite imagery data were also collected for Chiangrai, the province with the highest number of reported cases from 2003–2018. From 2003–2018, 103,345 cases of scrub typhus were reported with the number of reported cases increasing substantially over the observed period. There were more men than women, with agricultural workers the main occupational group affected. The majority of cases occurred in the 15–64 year old age group (72,144/99,543, 72%). Disease burden was greatest in the northern region, accounting for 53% of the total reported cases per year (mean). In the northern region, five provinces–Chiangrai, Chiangmai, Tak, Nan and Mae Hong Son–accounted for 84% (46,927/55,872) of the total cases from the northern region or 45% (46,927/103,345) of cases nationally. The majority of cases occurred from June to November but seasonality was less marked in the southern region. In Chiangrai province, elevation, rainfall, temperature, population size, habitat complexity and diversity of land cover contributed to scrub typhus incidence.

**Funding:** TW and NPJD are funded by the Wellcome Trust, UK, as part of the MORU Tropical Health Network institutional funding support. IE is funded by the Wellcome Trust, UK, as part of a Clinical Training Fellowship (grant number 105731/Z/14/Z). SM, RK and KC are supported by the French ANR FutureHealthSEA (ANR-17-CE35-0003-02) "Predictive scenarios of health in Southeast Asia: linking land use and climate changes to infectious diseases". The funders had no role in study design, data collection and analysis, decision to publish, or preparation of the manuscript.

**Competing interests:** We declare no competing interests.

## Interpretation

The burden of scrub typhus in Thailand is high with disease incidence rising significantly over the last two decades. However, disease burden is not uniform with northern provinces particularly affected. Agricultural activity along with geographical, meteorological and land cover factors are likely to contribute to disease incidence. Our report, along with existing epidemiological data, suggests that scrub typhus is the most clinically important rickettsial disease globally.

## Author summary

Scrub typhus, caused by the bacterium *Orientia tsutsugamushi*, is a major cause of fever in tropical and subtropical Asia. Symptoms can be similar to other common infections such as malaria, dengue, leptospirosis or typhoid, making it difficult to diagnose. Laboratory tests to diagnose scrub typhus are often unavailable, inaccurate, impractical, or too costly. Consequently, the true burden of scrub typhus disease remains unknown. Here, we collected data on the number of reported cases of scrub typhus in Thailand for 2003–2018 from the national surveillance system. There were 103,345 cases reported with incidence rising significantly over the study period. More men than women were affected, agricultural workers were the main occupational group (45%) and most cases (72%) were in adults of working age. The disease was seasonal with the cases mainly occurring during the rainy season from June to November. Five northern provinces were particularly affected with Chiangrai being the province with the highest number of cases. In Chiangrai, agricultural activity, elevation, rainfall, temperature and land cover factors contribute to disease burden. These results improve our understanding of the distribution and burden of scrub typhus in Thailand and help to identify factors that may contribute to disease incidence.

## Introduction

Scrub typhus in Asia, Australia and the Islands of the Indian and Pacific Oceans is a major cause of acute non-malarial febrile illness and a potentially severe but treatable zoonotic disease caused by the obligate intracellular Gram negative bacterium *Orientia tsutsugamushi* [1–4]. The disease is prevalent across an area of at least 13,000,000 km$^2$, encompassing large tracts of the Asia Pacific region, with emerging reports from the Middle East, South America, Africa and Europe suggesting the disease is more widespread than previously thought [5–9]. Median mortality rates have been estimated at 6.0% and 1.4% for untreated and treated disease, respectively, with a wide range of 0–70% depending on the study and the region [10, 11]. Transmission occurs through the bite of an infected larval stage (chigger) of the trombiculid mites [12]. Disease incidence may be related to seasonal variation in temperature, rainfall and humidity; socioeconomic factors, terrain, agricultural or other activities exposing humans to suitable habitat or transitional land cover type (mixed cropland and wild vegetation) favouring increased rodent and vector chigger density [13–16]. Although the disease is more prevalent in rural regions, cases from urban centres continue to be reported [17].

The incubation period ranges from 6 to 21 days and clinically, scrub typhus is characterised by fever, eschar, maculo-papular rash, headache, cough, nausea, vomiting, myalgia and

lymphadenopathy [18]. The eschar at the site of the mite bite is the most characteristic feature but is not always present, being dependent on the degree of pre-existing immunity to the infecting *Orientia tsutsugamushi* strain [19, 20]. Severe disease may present with complications affecting multiple organ systems [18]. The standard antibiotic treatment is doxycycline for 7 days with alternative options including tetracycline, chloramphenicol, azithromycin and rifampicin [21, 22]. In the majority of patients, fever will clear within 48–72 hours of treatment initiation. Reports of scrub typhus infections poorly responsive to treatment in northern Thailand and the lack of effective preventative measures, mainly vaccines with long-term multi-strain protection, remains a concern [23, 24].

Efforts to estimate the global burden of scrub typhus are hindered by limited data and difficulties surrounding diagnosis [11]. The frequently quoted estimate of one million cases annually lacks supportive evidence [25]. In Thailand, the first human case of scrub typhus was reported in 1952 although captured records from World War II suggest that cases among Japanese troops occurred from 1943 to 1944 [26, 27]. Subsequently, the disease has been shown to be highly prevalent and a major cause of acute febrile illness in the country [4, 28–30]. It is notifiable to the National Disease Surveillance System (R506), Bureau of Epidemiology (BoE), Department of Disease Control (DDC), Ministry of Public Health (MoPH), with government and private healthcare facilities in each province reporting to the public health office [31]. There are 76 provinces in Thailand (plus Bangkok, a special administrative area), grouped into 4 administrative regions (North, Northeast, Central and South), with each province being further divided into districts, sub-districts and villages. Within each province, the population is served by a central provincial hospital, district hospitals and primary care or health promoting units located in every sub-district. There has been an increase in notified cases reported from other countries with established passive notification systems such as South Korea, China and Bhutan, while results from Japan and more recently Taiwan suggest the disease incidence has stabilised [15, 32–37].

To date, limited data have been published on the national burden of scrub typhus in Thailand. In this study, we describe the burden of disease using national surveillance data from 2003–2018 and investigate the effect of geography, population, rainfall, temperature, habitat complexity and diversity of land use/land cover on disease incidence.

## Materials and methods

### Ethics statement

Publically available disease reporting data were obtained from the National Disease Surveillance System (R506), Bureau of Epidemiology (BoE), Department of Disease Control (DDC), Ministry of Public Health (MoPH). All data analysed were anonymised.

### Data collection

Detailed data for scrub typhus from 2003–2018 were obtained from the National Disease Surveillance System (R506) [31]. Brief annual summary reports from 1980–2002 were also available and collected. Probable and confirmed cases were reported according to the case definition, based on ICD-10: A75.3, and outlined in Table 1 below.

Cases were reported from governmental healthcare facilities including provincial hospitals, district hospitals and primary care units. Reporting from private healthcare facilities also occurred, albeit to a lesser extent. Reports for each year from 2003–2018 were obtained and collated into a single dataset. Information on the number of cases per province each month, population, annual incidence rate per 100,000 population (AIR), age groups, sex and occupation were collected. Additionally, less detailed information from annual summary reports

**Table 1. Scrub typhus reporting criteria.**

| | |
|---|---|
| **Clinical criteria** | Acute febrile illness and an eschar with at least one other symptom including:<br>- Headache<br>- Myalgia<br>- Arthralgia<br>- Ocular or orbital pain<br>- Petechial rash |
| **Laboratory criteria** | General findings suggestive of scrub typhus:<br>- Low white count<br>- Normal or low platelet count<br>Disease-specific:<br>- Detection of a four-fold rise in scrub typhus antibodies in paired sera by haemagglutination inhibition assay *or* antibodies detected at a cut-off titre of >1:1,280 in a single sample *or*,<br>Detection of scrub typhus IgM antibodies by ELISA *or*,<br>*Orientia tsutsugamushi* PCR *or* culture positive from blood |
| **Case classification**: | |
| • Probable case | Fulfil clinical criteria and has general laboratory findings suggestive of scrub typhus *or* epidemiological link to confirmed cases |
| • Confirmed case | Fulfil clinical and disease-specific laboratory criteria |

[ELISA–enzyme-linked immunosorbant assay, PCR–polymerase chain reaction]

from 1980–2002 were extracted and data collated into the main dataset. Regional administrative boundaries were obtained from the National Statistical Office, Ministry of Information and Communication Technology. As a case study, we obtained detailed reporting data for Chiangrai province from the Ministry of Public Health, the province with the highest number of reported cases from 2003–2018. Data included patient demographics, location and type of healthcare facilities, and residential data to the sub-district and village level. Administrative boundaries of the province including districts and sub-districts as well as population size were obtained from the Ministry of Natural Resources and Environment, Thailand (2012).

Average monthly temperature (˚C) and total monthly rainfall (mm) data for Chiangrai at a central representative weather station were obtained from the Thai Meteorological Department, Ministry of Information and Communication Technology. Land use information was obtained from the GlobCover 2009 with a resolution of 1 km (European Space Agency, http://due.esrin.esa.int/page_globcover.php, classification given in S1 Document and S1 Fig). Data were obtained at different administrative levels for Chiangrai province: scrub typhus incidence at village level, population size at sub-district level, rainfall at provincial level and land use at 1 km resolution. We integrated all data at the sub-district level, aggregating human cases obtained at village level and cropping raster data (land use) using sub-district boundaries.

## Statistical analyses

Proportions, percentages and averages (median and interquartile range [IQR] or mean and standard deviation [SD] as appropriate) were calculated controlling for any missing data. Seasonality was assessed by calculating proportions of cases (and 95% confidence intervals) reported during discrete time-periods and assessing for overlap as well as performing two-sample test of proportions. These descriptive analyses were performed using STATA 15 software (College Station, Texas, USA). Spatio-temporal analyses were conducted using R software (R Core Team, 2018) [38]:

i. Spatial autocorrelation of scrub typhus incidence and spatial interpolation using Gaussian process regression (kriging).

ii. Temporal incidence and temporal autocorrelation of scrub typhus cases.

iii. Spatiotemporal distribution of incidence using autocorrelation method, wavelet analysis and correlation between scrub typhus cases, rainfall and temperature.

iv. Analysed the links between scrub typhus cases and the agro-environmental data set using general linear modelling (GLM).

v. Performed general additive modelling (GAM), taking into account the spatiotemporal dynamics obtained from I, II and III and starting with the initial model that retained the potential explanatory factors of IV.

Further details on spatio-temporal analyses are outlined in S2 Document. Maps were drawn using mapchart.net ©, tmap and tmaptools [39, 40].

## Results

### National burden of scrub typhus in Thailand

From 2003–2018, there were a total of 103,345 cases of scrub typhus reported to the national disease surveillance system. During this period, the number of cases per year ranged from 2,928 in 2005 to 10,952 in 2013 and the annual incidence rate/100,000 population (AIR) ranged from 4.71 in 2005 to 17.09 in 2013. The median male to female ratio (IQR) was 1:0.74 (0.71–0.76) and the largest groups by occupation (median, IQR) were agricultural workers at 44.80% (42.65–45.88), labourers at 17.95% (15.77–19.22) and students at 14.83% (13.89–16.04). Data on the annual number of cases from 1980–2002 and incidence rate/100,000 population from 1985–2002 were also obtained. Cases per year ranged from 17 in 1980 to 5,094 in 2001 and AIR rose from 0.53 in 1985 to 8.2 in 2001. Sex and occupational groups were inconsistently reported from 1980–2002 but from the available data, the median (IQR) male to female ratio was 1:0.67 (0.56–0.76) and the percentage of agricultural workers was 51.35% (49.01–55.72). To visualise the trend over time, the total annual reported cases and AIR were plotted in (Fig 1A and 1B, respectively).

Age group data were available from 2004–2018 (S2 Fig). The age groups with the highest number of cases per year (mean and SD) were 45–54 years at 1,126 (519) cases, 35–44 years at 1,086 (402) cases and 25–34 years at 972 (303) cases. The majority of cases reported from 2004–2018 were in the 15–64 year old age bracket (72,144/99,543, 72.48%). Indeed, the proportion of cases age <15 years old has been gradually falling from 825/3,290 (25.08%) of cases in 2004 to 1,396/9,756 (14.31%) of cases in 2018.

Cases per month from 2003–2018 were plotted in Fig 2. The months with the highest average number of cases (mean, SD) were October, July and September at 809 (361), 772 (315), and 757 (342) cases, respectively. The majority occurred between June and November, corresponding to the rainy and early cool or dry seasons in Thailand. The proportion of cases (95% CI) from June to November and December to May were calculated: 0.678 (0.675–0.681) and 0.322 (0.319–0.325), p<0.001.

### Regional and provincial burden of scrub typhus

From 2003–2018, the greatest burden of disease was in the northern region which accounted for approximately half the total cases (see Fig 3 for map of administrative regions). The percentage of cases (mean, SD) from the north per year was 52.55% (7.49%) despite the region contributing only 18.60% (SD 0.36%) of the total population during this period. This is reflected in the mean AIR of 29.40 (SD 13.39) cases for the study period. The mean percentage of cases per year (SD) and mean AIR (SD) from 2003–2018 for the northeast, south and central

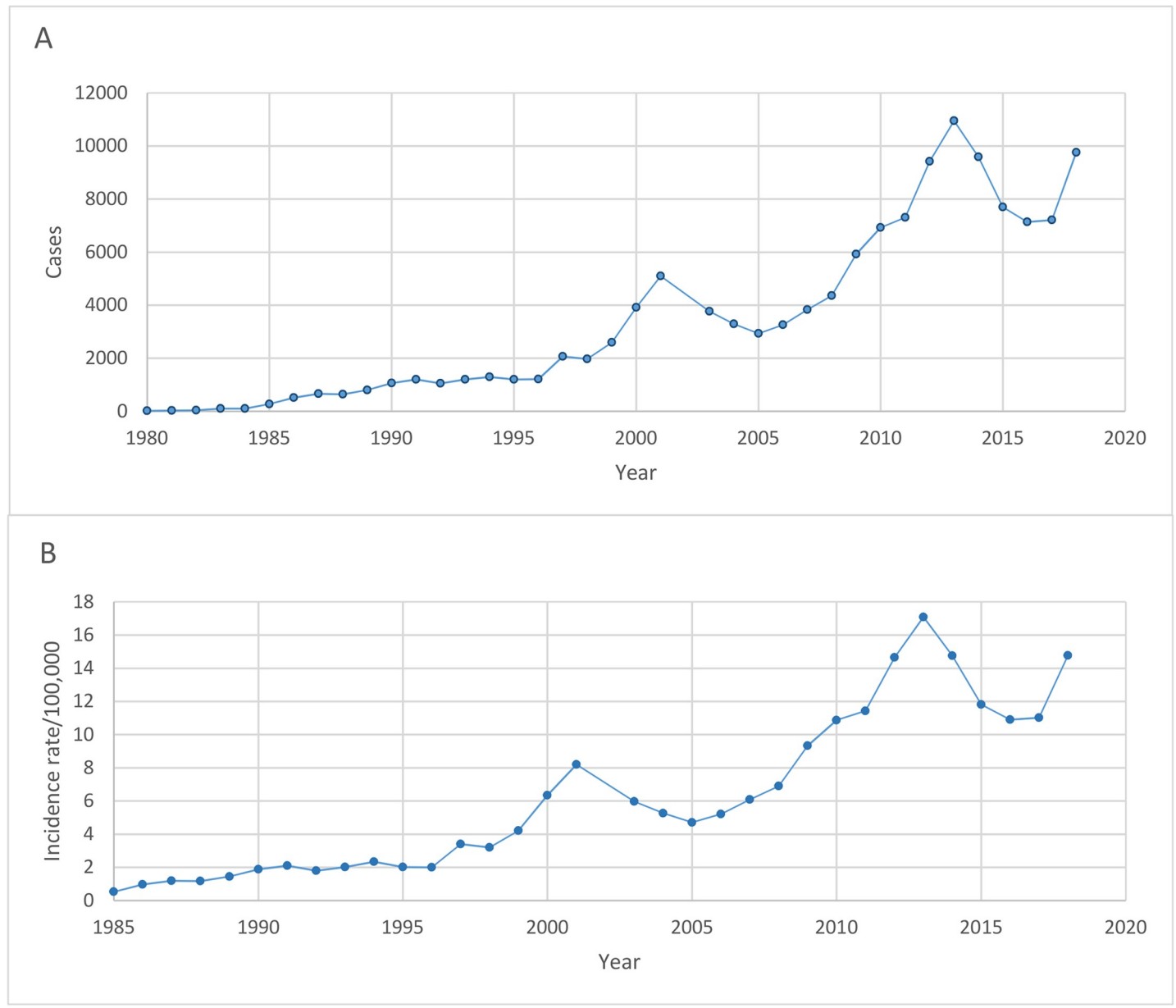

**Fig 1.** Trend in the annual number of reported scrub typhus cases (A) from 1980–2018 and annual incidence rate per 100,000 population (B) from 1985–2018.

regions were 30.92% (6.97%) and 9.01 (3.94) cases, 12.40% (2.54%) and 8.94 (4.04) cases, and 4.12% (2.49%) and 0.99 (0.30) cases, respectively.

Detailed occupational data to the regional and provincial level were available from 2006–2018. In the North, agricultural workers made up the largest percentage of cases (median, IQR) at 40.35% (39.72–42.29), followed by labourers at 19.36% (18.79–20.94) and students at 16.45% (15.10–19.10). In the Northeast, there was a larger percentage of agricultural workers at 57.78% (52.82–60.93), followed by labourers and students at 11.31% (10.36–12.57) and 10.47% (9.56–13.79), respectively. A similar pattern was seen in the South, albeit to a lesser extent, with agricultural workers at 33.33% (31.67–33.70), labourers at 22.72% (20.74–25.37) and students at 15.71% (13.68–17.65). A difference was seen in the central region where

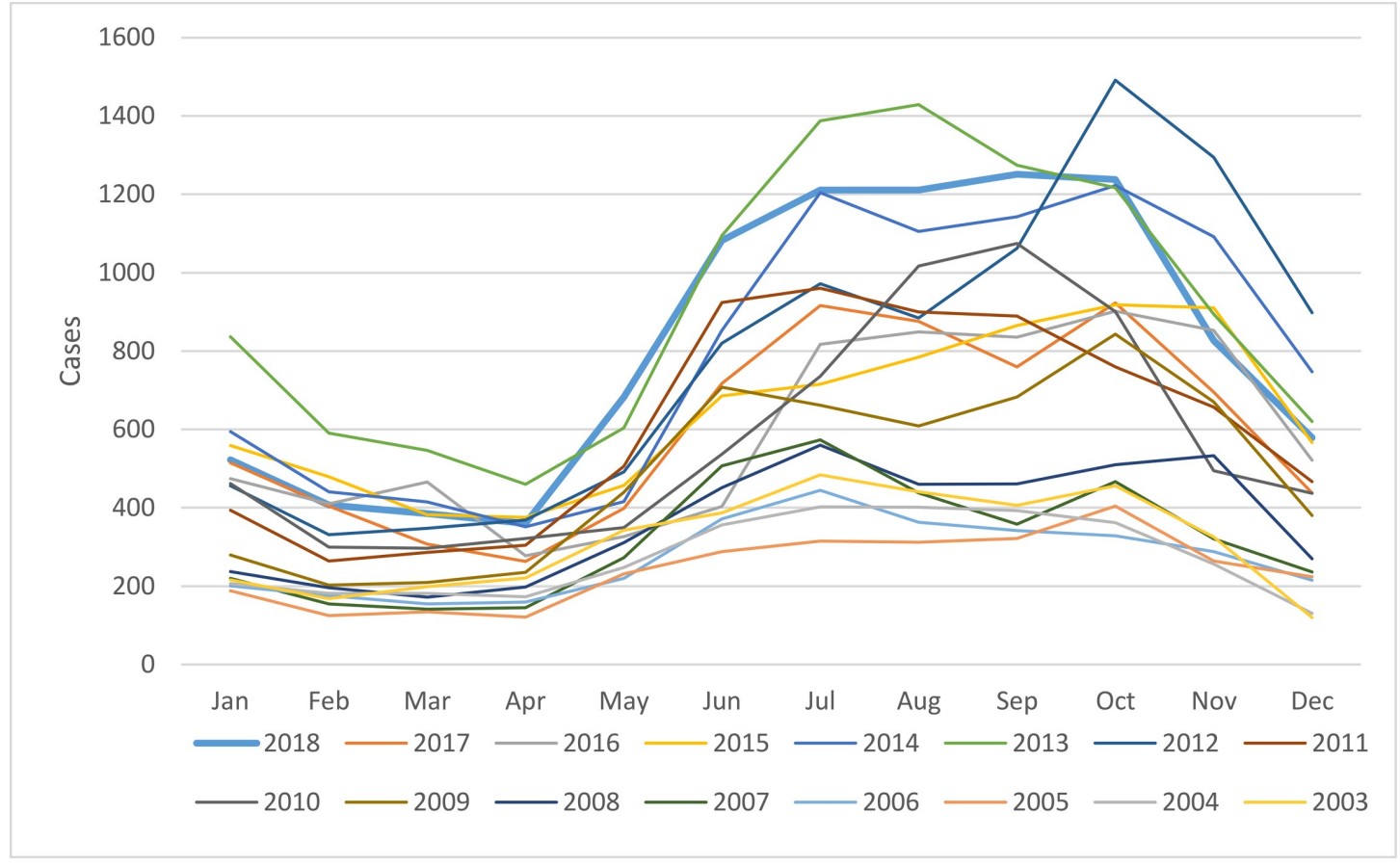

**Fig 2. Scrub typhus cases by month from 2003–2018.**

labourers made up 40.59% (38.06–42.51) of the cases, followed by students at 18.54% (16.48–21.36) and agricultural workers at 12.57% (10.71–13.95).

Regional differences were also observed in the seasonality of cases reported from 2003–2018 as shown in Fig 4. In all four regions, average monthly reported cases were at their lowest from February to April before rising in May. In the North and Northeast, cases peaked in July and October, respectively, before falling in November and December, reaching a trough from February to April. In the southern and central regions, the decline in cases at the end of the year was interrupted by another peak in January prior to falling to a nadir in April. Seasonality of scrub typhus cases was less marked in the southern region when compared to other regions. From 2003–2018, the proportion of cases (95% CI) from June to November and from December to May were calculated for the northern, northeastern, central and southern regions: 0.687 (0.683–0.691):0.313 (0.309–0.317), p<0.001; 0.726 (0.721–0.731):0.274 (0.269–0.279), p<0.001; 0.618 (0.601–0.634):0.382 (0.366–0.399), p<0.001; and 0.538 (0.530–0.547):0.462 (0.453–0.470), p<0.001, respectively.

At the provincial level, the greatest burden of scrub typhus was in the northern provinces (Fig 5). Indeed, the five provinces with the highest average (mean, SD) cases per year from 2003–2018 were Chiangrai– 716 (378), Chiangmai– 684 (435), Tak– 609 (361), Nan– 488 (292) and Mae Hong Son– 437 (221). These five provinces accounted for 46,927 reported cases over the 16 year study period which represents 46,927/55,872 (83.99%) of cases from the northern region, or 46,927/103,345 (45.41%) of cases nationally. The highest AIRs (mean, SD)

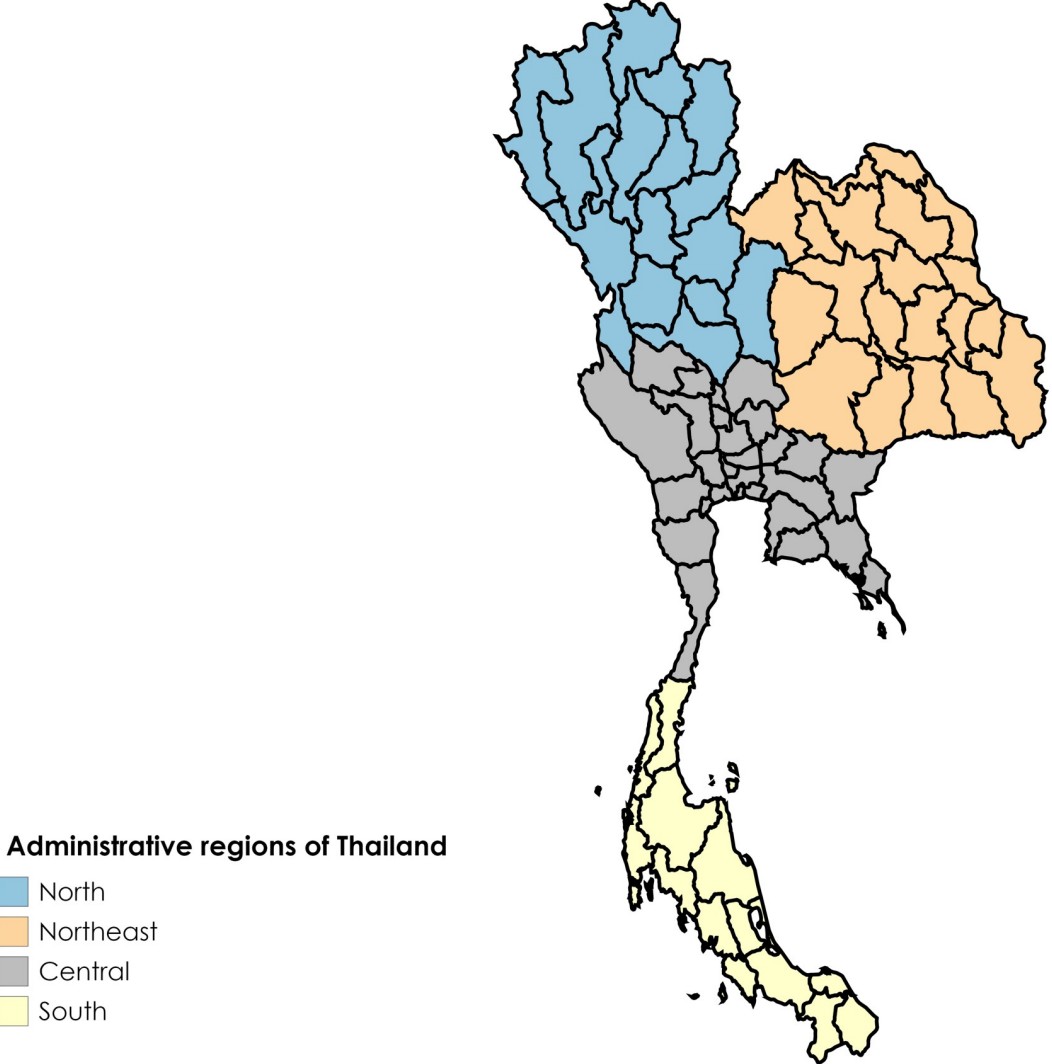

**Fig 3. Administrative regions of Thailand [National Statistical Office, Ministry of Information and Communication Technology; reprinted from mapchart.net under a CC BY license, with permission from Minas Giannekas original copyright 2019].**

from 2003–2018 were seen in Mae Hong Son– 173.14 (85.62), Tak– 111.40 (64.54), Nan– 102.07 (61.20), Chiangrai– 58.61 (30.80) and Phang Nga—47.98 (27.85) provinces. Of these, only Phang Nga province (southern region) is outside of northern Thailand.

## Single province case study–Chiangrai province

Chiangrai province was chosen as it represented the province with the highest average number of scrub typhus cases per year along with the 4th highest disease incidence rate per 100,000 population. The province is the northernmost province in Thailand and is divided into 18 districts, 124 sub-districts and 1,816 villages. From 2003–2018, there were a total of 11,444 scrub typhus cases reported to the provincial public health office. The male to female ratio was 1:0.81. The age groups with the highest mean (SD) number of cases per year were 45–54 years at 115 (65) cases, 35–44 years at 109 (55) cases and 25–34 years at 91 (44) cases while the percentage of paediatric cases (age <15 years old) has fallen from 34.01% in 2003 to 21.55% in

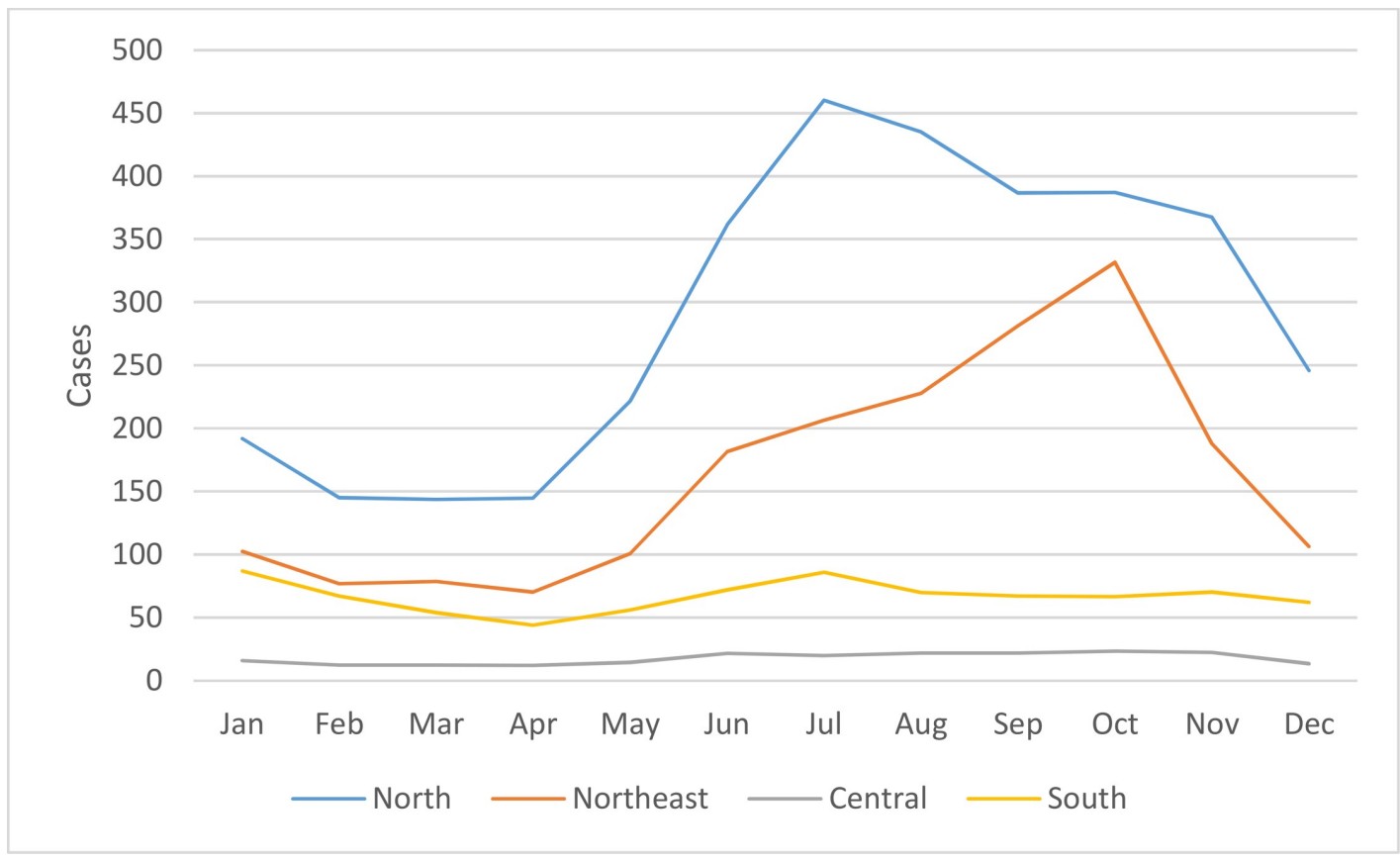

**Fig 4. Seasonality of scrub typhus cases by month per region from 2003–2018.**

2018. The months with the highest average number of cases (mean, SD) were July, August and June at 110 (73), 95 (49), and 90 (55) cases, respectively. The proportion of cases (95% CI) from June to November and December to May for Chiangrai province from 2003–2018 were calculated: 0.716 (0.708–0.7205) and 0.284 (0.275–0.292), p<0.001, respectively.

The main occupations reported were agricultural workers at 35.63%, labourers at 22.76% and students at 20.06%. The majority of scrub typhus patients attended district hospitals 75.26% or the provincial hospital 22.51% while a very small number of cases were reported from private health facilities 1.69%. The majority of patients (66.89%) were managed as outpatients.

From 2003–2018, Mae Fah Luang district had the highest total number of reported cases with 2,413 cases, followed by Mae Suai district with 2,014 cases and Mueang district with 1,431 cases. These 3 districts, located mainly to the west of the province, contributed to around half the total provincial burden over the 16 year period (5,858/11,444–51.19%) despite contributing a combined population of 423,063/1,326,865 (31.88%) of the entire province. Highest average AIR were seen in Mae Fah Luang (212.84), Mae Suai (131.45) and Wiang Pa Pao (96.25) districts.

**I. Spatial autocorrelation, semi-variogram and kriging interpolation of scrub typhus cases.** A weak spatial autocorrelation was observed when pooling scrub typhus cases for 2003–2018 for Chiangrai province but was no longer significant after 30 km (S3 Fig). The spatial distribution of the scrub typhus cases among sub-districts was analysed using semi-variogram analysis and depicted in Fig 6B. The kriging interpolation, using the results of the semi-

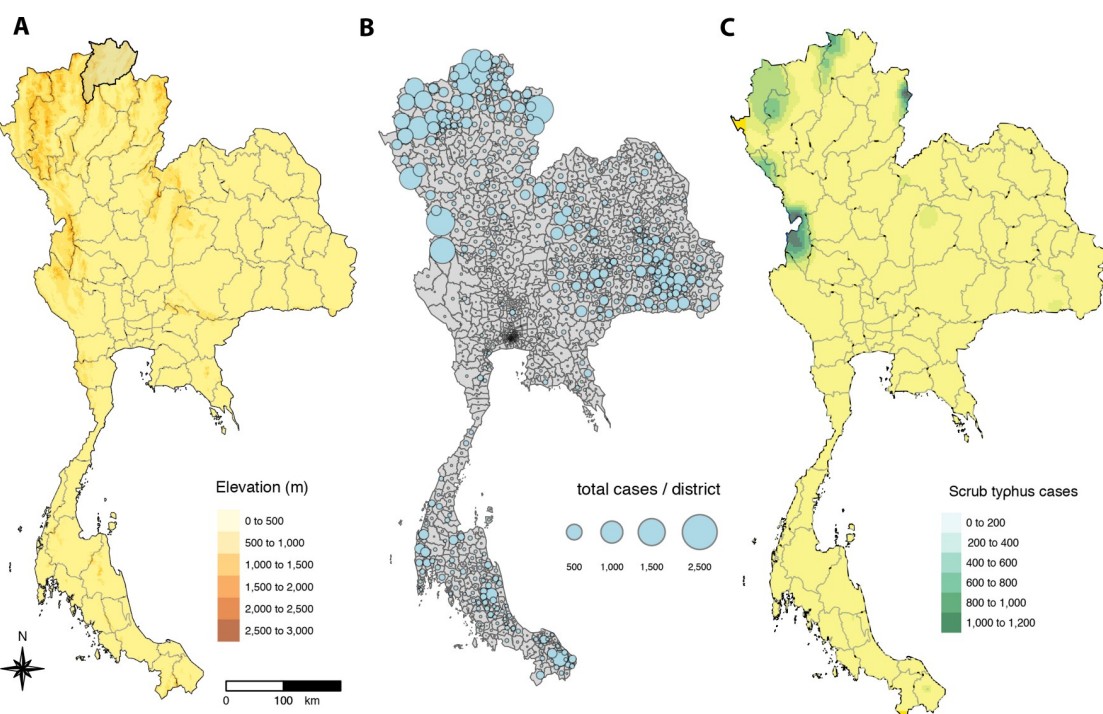

**Fig 5.** Map of Thailand depicting: (A) the geography with elevation, provincial boundaries and location of Chiangrai province; (B) total scrub typhus cases from 2003–2018 per district; and (C) interpolation of scrub typhus cases at sub-district level by kriging using semi-variogram based on the centroids of geographical coordinates of each sub-district (for clarity, only provincial boundaries are shown) [created using tmap and tmaptools in R software–R Core Team 2018 [38–40]].

variogram analysis, is represented in Fig 6C. Spatial interpolation of scrub typhus cases showed hotspots of cases in high elevation area of the province (Fig 6A and 6C).

**II. Time series analysis of scrub typhus cases, rainfall and temperature.** There was an increasing trend in reported scrub typhus cases per month for Chiangrai from 2003 to 2018

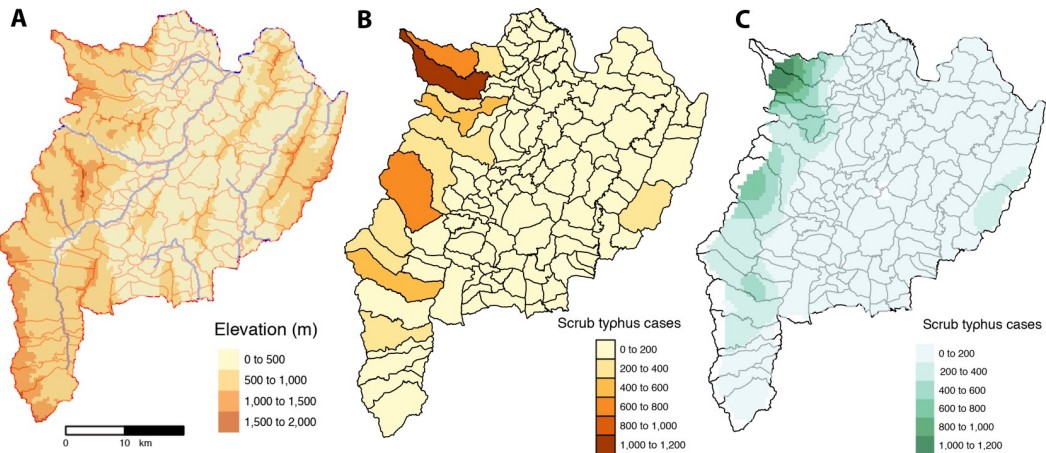

**Fig 6.** Chiangrai province with: (A) the geography with elevation, main rivers and sub-district boundaries; (B) scrub typhus cases from 2003–2018 per sub-district; and (C) interpolation of scrub typhus cases at sub-district level by kriging using semi-variogram based on the centroids of geographical coordinates of each sub-district [created using tmap and tmaptools in R software–R Core Team 2018 [38–40]].

### A    Scrub typhus cases

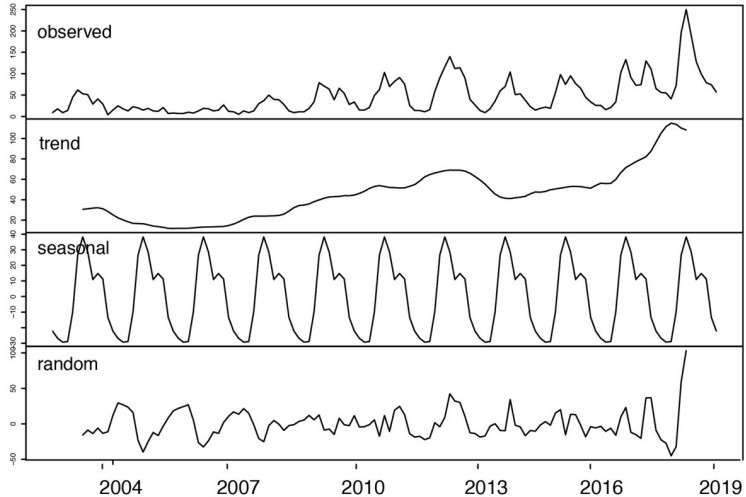

### B    Rainfall

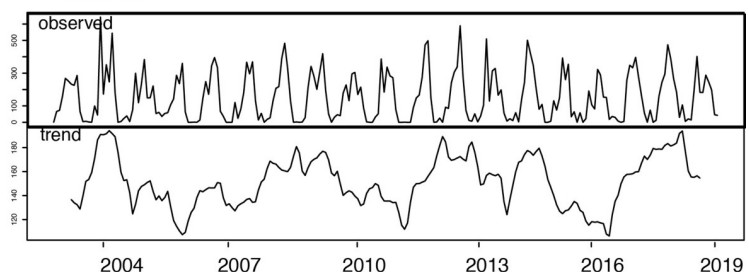

### C    Temperature

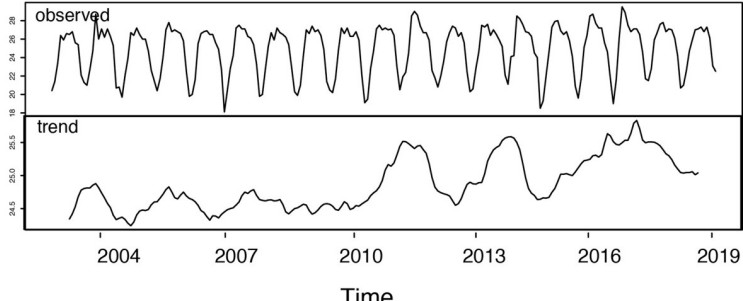

**Fig 7.** Time series analysis of (A) reported monthly scrub typhus cases, (B) total monthly rainfall in mm and (C) average monthly temperature in ˚C for Chiangrai province.

with a marked increase after 2010 (Fig 7). There was a strong seasonal pattern with large numbers of scrub typhus cases during the rainy season. The wavelet analysis (S4 Fig) confirmed a seasonal pattern starting in 2008. A strong seasonal pattern for both rainfall and temperature was also observed (Fig 7). A slight increase in average temperature from an average of 24.5˚C in 2003 to 25.5˚C in 2018 was observed, coinciding with the increase in reported cases. Wavelet power spectrum (S4 Fig) confirmed the above time series analyses by revealing significant 12-month periodicity over the entire study period.

**Table 2. Results of the best global GLM with binomial negative link function (p<0.001), with theta = 2.12 (standard deviation = 0.026), explaining the number of scrub typhus cases in Chiangrai province from 2003–2018 by sub-district and by month with explanatory geographical and meteorological variables in the initial model.** Estimate and standard deviation (SD) were given for each selected explanatory variables, with P value and VIF (Variance Inflation Factor). For the best selected model the log likelihood = -8925.35 with degree of freedom (DF) = 12, null deviance = 3497.2, $R^2$ estimated by maximum likelihood ($R^2$ML) = 0.25, $w_r$ = 0.36 and Akaike Information Criteria (AIC) = 17875 (see S6 Fig).

| Explanatory variables | Estimate (SD) | P value | VIF* | Odds ratio (CI 2.5%- 97.5%) |
|---|---|---|---|---|
| Population size | 0.00002 (0.000001) | <0.0001 | 1.45 | 1.45 (1.38–1.53) |
| Total monthly rainfall | 0.001 (0.0001) | <0.0001 | 1.53 | 1.02 (1.01–1.02) |
| Average monthly temperature | 0.025 (0.005) | <0.0001 | 1.53 | 1.87 (1.44–2.42) |
| Forest cover (%) | -4.19 (0.53) | <0.0001 | 1.44 | 0.81 (0.77–0.86) |
| Forest open cover (%) | 0.45 (0.22) | 0.046 | 22.55 | 1.56 (1.01–2.41) |
| Grassland open cover (%) | -0.0004(0.0001) | <0.0001 | 4.91 | 0.84 (0.78–0.90) |
| Mosaic habitat cover (%) | -0.78 (0.20) | 0.0001 | 7.60 | 0.79 (0.70–0.89) |
| Rain-fed land cover (%) | -0.73 (0.18) | <0.0001 | 8.89 | 0.80 (0.72–0.89) |
| Flooded-irrigated land cover (%) | -0.99 (0.32) | 0.002 | 2.09 | 0.88 (0.81–0.95) |
| Habitat complexity | 0.14 (0.013) | <0.0001 | 5.11 | 1.51 (1.40–1.62) |
| Habitat fragmentation | -0.02 (0.004) | <0.0001 | 1.28 | 0.80 (0.74–0.87) |

(*value of VIF > 10 for continuous variables may indicate problems with collinearity)

**III. Cross temporal correlation analysis.** Cross-correlation analysis among pairs of uni-variate series revealed moderate correlation between rainfall and scrub typhus cases (R = 0.46) with a lag time of one month and moderate correlation between temperature and scrub typhus cases (R = 0.55) with a lag time of two months (S5 Fig).

**IV. Association between scrub typhus cases and explanatory variables using General Linear Modelling (GLM).** A GLM analysis was performed using the following explanatory variables: habitat complexity, habitat fragmentation, forest cover (%), forest open cover (%), grassland open cover (%), mosaic habitat cover (%), rain-fed cropland cover (%), flooded-irrigated land cover (%), population size, rainfall and temperature (S2 Document). The top best model selected showed that scrub typhus cases per sub-district per month were significantly associated with all variables included in the initial model supported by an Akaike weights ($w_r$) value of 0.36 (S2 Document, Table 2).

Multi-collinearity assessed using Variance Inflation Factor (VIF) showed that most VIF values were inferior to 10, suggesting lack of collinearity, except for forest open cover with a VIF value of 22.55, suggesting high collinearity with other land use characteristics. The percentage of deviance explained by this model was estimated by maximum likelihood $R^2$ to 0.25 indicating a good prediction of this best model.

A positive association was found between population size and scrub typhus cases. The number of cases were also positively correlated with temperature and rainfall. Higher numbers of scrub typhus cases were found associated with habitat complexity characterised by large cover of open forested habitat. Lower numbers of scrub typhus cases were associated with fragmented habitat characterised by mosaic habitat or open grassland habitat. Negative associations were also observed between scrub typhus cases and forested, rain-fed and flooded-irrigated lands. The above relationships are depicted in S6 Fig.

**V. Association between scrub typhus cases and explanatory variables using General Additive Modelling (GAM).** An initial GAM model was developed that took into account the spatiotemporal dynamics of scrub typhus cases, the temporal dynamics of rainfall and temperature, and the potential explanatory variables selected by the GLM, with the exception of open forested habitats (due to high VIF value), using a negative binomial function (with theta

**Table 3. Results of general additive modelling (GAM) explaining the number of cases of scrub typhus per sub-district in Chiangrai province using a negative binomial link (theta = 2.12), with approximate significance of smooth terms.** For the best selected model, the deviance explained = 50.4%, $R^2$ = 0.37, maximum likelihood = 8868.2, AIC = 17689.8 (see Fig 8).

| Explanatory variables | Estimated degrees of freedom | Reference DF | Chi square | P value |
|---|---|---|---|---|
| Longitude and latitude | 16.71 | 29 | 117.1 | <0.0001 |
| Total monthly rainfall | 11.36 | 24 | 155.5 | <0.0001 |
| Average monthly temperature | 0.97 | 20 | 1.9 | 0.10 |
| Population size | 2.89 | 9 | 87.2 | <0.0001 |
| Habitat complexity | 1.14 | 9 | 6.6 | 0.002 |
| Rain-fed land cover (%) | 1.76 | 9 | 8.8 | 0.001 |
| Mosaic habitat cover (%) | 0.86 | 4 | 3.3 | 0.027 |
| Flooded-irrigated land cover (%) | 0.84 | 9 | 3.3 | 0.024 |

estimated as above). The best GAM model selected using Akaike Information Criteria (AIC) is shown in Table 3 and Fig 8, which did not include the variables habitat fragmentation and grassland open cover (%).

The model explained 50.4% of the deviance ($R^2$ = 0.37) with a better performance than the above best GLM. The best GAM model confirmed the different results obtained above with a significant influence of geography (matrix of geographic coordinates of sub-districts), rainfall, temperature, population, habitat complexity, mosaic habitat, rain-fed and flooded-irrigated lands (Table 3, Fig 8).

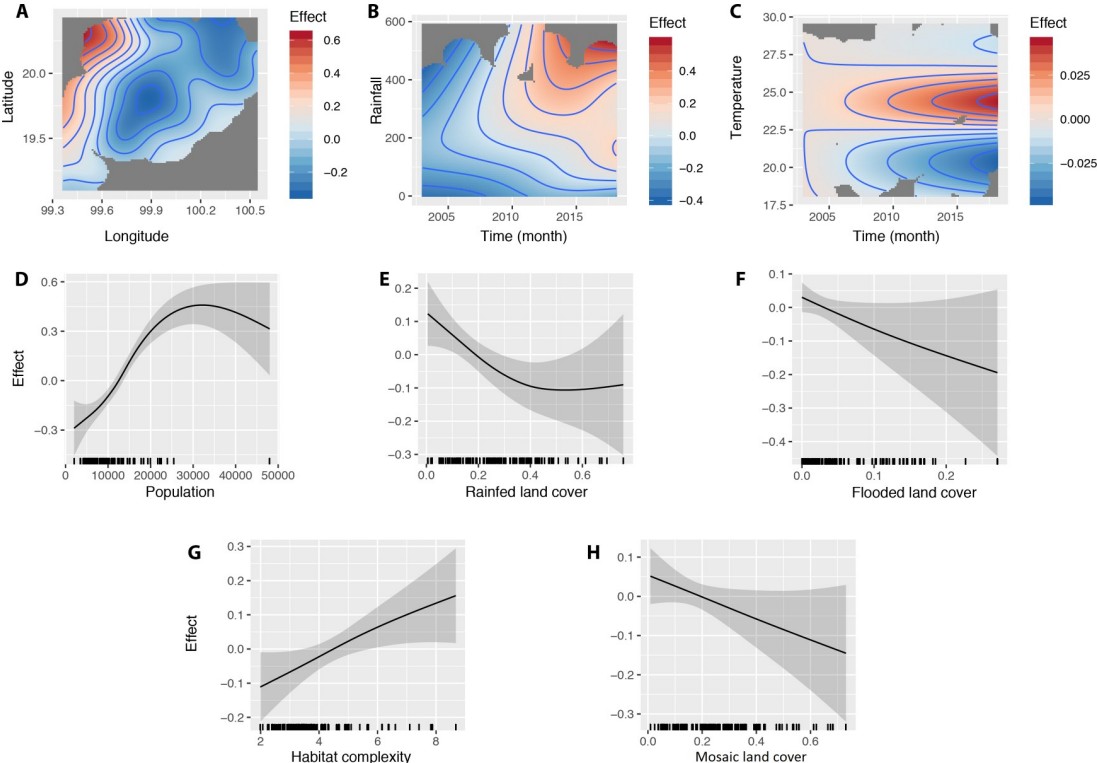

**Fig 8. Results of the best General Additive Modeling (GAM) explaining the number of scrub typhus cases in Chiangrai province over 2003–2018 by sub-district and by month, using a binomial negative link function (theta = 2.12).** The smoothed variables selected in the best GAM were (A) the geographical distribution of sub-district (longitude and latitude of the centroid), (B) total monthly rainfall (mm), (C) temperature (˚C), (D) population size, (E) rain-fed land cover, (F) flooded land cover, (G) habitat complexity and (H) mosaic land cover.

## Discussion

The results of this study reveal a high burden of scrub typhus in Thailand that has increased significantly over the last two decades. Although cases have been reported from all 77 provinces from 2003–2018, the burden of disease is not uniform with a handful of northern provinces particularly affected. We have also shown that within Chiangrai province, scrub typhus is highly prevalent in the rural and mountainous areas to the western, northwestern and, to a lesser extent, eastern parts of the province.

Our results from Thailand are not unexpected when compared to other countries with reported surveillance data. There have been increases in scrub typhus cases reported in South Korea (2001–2013, 70,914 cases), China (2006–2014, 54,558 cases) and Bhutan (2011–2014, 778 cases), but not Japan (2006–2017, 4,185 cases) or Taiwan (2003–2018, 6,746 cases) where the burden has remained stable [32–36, 41]. Similar to Thailand, there were more cases in men than women in Japan and Taiwan but the opposite was seen in China and South Korea. In Thailand, the disease burden was highest in adults of working age while the proportion of children affected continues to fall, reflecting the accelerated increase in disease burden in adults compared to children. Scrub typhus was more common in older age groups in the other five countries, particularly Japan and South Korea with its ageing farming communities [42]. Unsurprisingly, high disease incidence among farmers in China, eastern Taiwan and South Korea were also described and in conjunction with our data, highlights the ongoing risk of disease in agricultural workers [32–34]. Increasing urbanisation of scrub typhus has recently been reported from Seoul, South Korea, which may either reflect increased human exposure to rural habitats as urban centres expand or the increase in development of parkland and gardens within municipal boundaries [17].

Seasonality of scrub typhus was observed in China, Taiwan, South Korea and Japan. In Japan, the disease incidence peaked in November and December with a smaller peak in May and June [36]. South Korea reported a disease peak from October to December with lower latitudes associated with a later peak in incidence [15, 32]. In Taiwan, the disease incidence was biphasic with the main peak in June and July and a second smaller peak in September and October [41]. In mainland China, intra-country variation in seasonality was observed with cases in the northern provinces occurring mainly in autumn and early winter while in the southern provinces, the scrub typhus "season" began in spring and extended to the end of the year [33]. The disease pattern in southern China and Taiwan most closely resemble the results from northern and northeastern Thailand which is not unexpected given their similar latitudes. However, our study clearly demonstrates the change in disease seasonality in Thailand as latitude decreases. This effect was more pronounced than the study from South Korea and may reflect Thailand's proximity to the equator along with its greater latitude range [15]. In southern Thailand, there are two main seasons (wet and dry) and the temperature range is narrower than northern Thailand. Rainfall tend to be higher in the South, with the southwest monsoon affecting the west coast of the peninsula earlier in the season (April to October) while the east coast tends to have higher rainfall later (September to December). In higher latitudes, winters are frequently cold and summers hot, the temperature range affecting the population of vector chiggers [43]. This is demonstrated in Japan, where monthly incidence is more evenly distributed moving southwards [44]. In tropical regions, temperature varies less and here rainfall appears to influence chigger abundance and thus human risk [45, 46].

In Chiangrai province, modelling revealed the partial contribution of geography (elevated or mountainous areas), rainfall, temperature, population size, habitat complexity and diversity of land cover to the number of reported scrub typhus cases per sub-district. The lag time of 2 months for temperature and 1 month for rainfall in the cross correlation analysis could be

explained by agricultural activity, such as the clearing of scrub or new vegetation prior to planting in June/July, or chigger activity during the rainy season. Habitats characterised by growth of new or transitional vegetation and associated with the presence of infected rodents and chigger mites are important ecological factors [27, 47]. These habitats are diverse and include virgin forests, gardens, fringe habitats or scrubland, water-meadows, beachheads, rice paddies, bamboo patches and oil palm or rubber plantations, many of which are present in Chiangrai. Although there was a significant effect of population on disease incidence, the proportion of the population exposed to the habitats described is likely to be more predictive but will be harder to obtain. The association of elevation and scrub typhus incidence in our study reflect findings from southern China [16]. This may reflect the reduced accessibility and greater habitat complexity of these regions. In addition, an increase in average monthly temperature from 24.5°C in 2003 to 25.5°C in 2018 correlated with an increase in scrub typhus incidence. This finding is also similar to reports from southern China and together, are suggestive of the influence of climate change on disease burden [16, 48, 49].

Diagnostic capabilities for scrub typhus at district hospitals are limited and most provincial hospitals are not able to perform confirmatory assays in-house, requiring the aid of reference laboratories for confirmation of diagnosis [50]. Antibody detection-based rapid diagnostic tests (RDTs) for scrub typhus have been introduced (e.g. since 2008 in Chiangrai province) but are sub-optimal [51]. The majority of scrub typhus cases reported in Thailand are probable cases based on the clinical criteria of fever and eschar (with or without positive RDT results). This combination of clinical findings has a sensitivity and specificity of 47% and 81% in adults and 60% and 92% in children for scrub typhus in Chiangrai [52, 53]. Despite this weakness, a scrub typhus surveillance system based on a similar clinical criteria for probable cases can play a valuable role in allowing the disease burden to be extrapolated, particularly in regions where scrub typhus is the dominant eschar-associated disease (alternatives include spotted fever rickettsial infections or necrotic spider bite wounds).

Other limitations include the passive nature of the surveillance system and the effect of healthcare-seeking behaviour on the data used in this study. Passive surveillance systems are reliant on facilities and healthcare and public health staff for data completeness and quality. Although reporting data from private healthcare facilities are included in the analysis, it is unclear whether cases are consistently reported from the private sector. These limitations suggest that the burden of scrub typhus reported in this study is likely to be a gross underestimate of the true disease burden. Additionally, we were unable to measure the influence of other factors including the increase in disease awareness, availability of diagnostic tests, agricultural activity, human behaviour, and chigger abundance and activity in influencing the impressive rise in reported cases since 1980. Finally, although the data on patients' villages of residence were collected, accurate geolocation data proved particularly difficult to obtain which limited the utility of this high-resolution data. However, the data will allow public awareness and engagement programmes to target villages and sub-districts with the highest burden of scrub typhus. Paradoxically, despite the high incidence of scrub typhus in these areas, awareness among community health workers remains low and is almost completely absent in villagers.

In this report, we have estimated burden of scrub typhus in Thailand from national surveillance data for 2003–2018. Considering that currently available epidemiological data are limited, these results should contribute significantly towards our understanding of the burden of disease both locally and globally. Additional work on the epidemiology of scrub typhus and the development of accurate and cost-effective diagnostic tests should be prioritised. The single province case study has shown how geography, rainfall, temperature and landscape complexity may partly contribute to disease incidence. Further data collection and analyses to investigate the impact of geographical and meteorological factors on disease incidence for the remaining

provinces are ongoing. The burden of scrub typhus from Thailand and other countries within and beyond the traditional endemic region suggest it is the most clinically important rickettsial disease globally.

## Supporting information

**S1 Document. Land use–land cover classification for Chiangrai province.** GlobCover 2009 with a resolution of 1 km was used (http://due.esrin.esa.int/page_globcover.php). (DOCX)

**S2 Document. Details of spatio-temporal analyses performed on reported scrub typhus data, geographical data and meteorological data from Chiangrai province.** (DOCX)

**S3 Document. STROBE checklist.** (DOCX)

**S1 Fig. Classification of land use–land cover for Chiangrai province.** GlobCover 2009 satellite imagery was used. (TIF)

**S2 Fig. Age-group data of reported scrub typhus cases for Thailand from 2004–2018.** (TIF)

**S3 Fig. Spatial autocorrelation (distance in km) of reported scrub typhus cases for Chiangrai province from 2003–2018.** (TIF)

**S4 Fig.** Wavelet analysis for Chiangrai province depicting (A) seasonal pattern of scrub typhus cases from 2008 onwards, (B) significant 12-month periodicity of total monthly rainfall from 2003–2018 and (C) significant 12-month periodicity of average monthly temperature from 2003–2018. (TIF)

**S5 Fig. Cross temporal correlation analysis for scrub typhus cases, total monthly rainfall (mm) and average monthly temperature (˚C) for Chiangrai province.** Significant correlation was seen between rainfall and scrub typhus cases (R = 0.46, A) with a lag time of one month along with temperature and scrub typhus cases (R = 0.55, B) with a lag time of two months. (TIF)

**S6 Fig. Results of the best General Linear Modeling (GLM) explaining the number of scrub typhus cases in Chiangrai province over 2003–2018 by sub-district and by month, using a binomial negative link function (with theta = 2.12).** The smoothed variables selected in the best GLM were (A) habitat complexity, (B) population, (C) habitat fragmentation, (D) forest cover, (E) forest open cover, (F) grassland open cover, (G) mosaic habitat cover, (H) rain-fed land cover, (I) flooded-irrigated land cover, (J) average monthly temperature in ˚C and (K) total monthly rainfall in mm. (TIF)

## Acknowledgments

We would like to thank Dr Pawinee Doung-ngern and other staff at the Communicable Disease Unit, Bureau of Epidemiology, Department of Disease Control, Ministry of Public Health

for their help with data collection and permission to utilise the data. We thank staff at the Ministry of Natural Resources and Environment and Thai Meteorological Department, Ministry of Information and Communication Technology for their help with data collection. Additionally, we thank Nidanuch Tasak and Piangnet Jaiboon for their role in obtaining data for Chiangrai province.

## Author Contributions

**Conceptualization:** Tri Wangrangsimakul, Ivo Elliott, Kittipong Chaisiri, Nicholas P. J. Day, Serge Morand.

**Data curation:** Tri Wangrangsimakul, Serge Morand.

**Formal analysis:** Tri Wangrangsimakul, Serge Morand.

**Funding acquisition:** Nicholas P. J. Day, Serge Morand.

**Investigation:** Tri Wangrangsimakul, Ivo Elliott, Supalert Nedsuwan, Rawadee Kumlert, Soawapak Hinjoy, Kittipong Chaisiri, Serge Morand.

**Methodology:** Tri Wangrangsimakul, Ivo Elliott, Kittipong Chaisiri, Serge Morand.

**Project administration:** Tri Wangrangsimakul, Supalert Nedsuwan, Rawadee Kumlert, Soawapak Hinjoy, Nicholas P. J. Day, Serge Morand.

**Resources:** Tri Wangrangsimakul, Supalert Nedsuwan, Soawapak Hinjoy, Nicholas P. J. Day, Serge Morand.

**Software:** Tri Wangrangsimakul, Serge Morand.

**Supervision:** Nicholas P. J. Day, Serge Morand.

**Validation:** Tri Wangrangsimakul, Nicholas P. J. Day, Serge Morand.

**Visualization:** Tri Wangrangsimakul, Serge Morand.

**Writing – original draft:** Tri Wangrangsimakul, Ivo Elliott, Kittipong Chaisiri, Serge Morand.

**Writing – review & editing:** Tri Wangrangsimakul, Ivo Elliott, Supalert Nedsuwan, Rawadee Kumlert, Soawapak Hinjoy, Kittipong Chaisiri, Nicholas P. J. Day, Serge Morand.

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
