## [Decision Letter · Decision Letter 0]

26 Jan 2020

Dear Dr. Wangrangsimakul,

Thank you very much for submitting your manuscript "The estimated burden of scrub typhus in Thailand from national surveillance data (2003-2018)" for consideration at PLOS Neglected Tropical Diseases. As with all papers reviewed by the journal, your manuscript was reviewed by members of the editorial board and by several independent reviewers. The reviewers appreciated the attention to an important topic. Based on the reviews, we are likely to accept this manuscript for publication, providing that you modify the manuscript according to the review recommendations. 

Thank you for submitting your manuscript to PLoS NTD. We request that you kindly respond to the reviewers comments and resubmit a revised manuscript

Sincerely,

Husain Poonawala

Guest Editor

Hélène Carabin

Deputy Editor

Thank you for submitting your manuscript to PLoS NTD. We request that you kindly respond to the reviewers comments and resubmit a revised manuscript

Reviewer's Responses to Questions

**Key Review Criteria Required for Acceptance?**

**Methods**

-Are the objectives of the study clearly articulated with a clear testable hypothesis stated?

-Is the study design appropriate to address the stated objectives?

-Is the population clearly described and appropriate for the hypothesis being tested?

-Is the sample size sufficient to ensure adequate power to address the hypothesis being tested?

-Were correct statistical analysis used to support conclusions?

-Are there concerns about ethical or regulatory requirements being met?

Reviewer #1: The objectives of the study are clearly articulated, and study design is appropriate to address the objectives. The population is clearly described in appropriate and there are no ethical concerns.

Reviewer #2: The methods carried out for the investigation of the burden of scrub typhus in Thailand are clearly described and appropriate to the objective of the study.

**Results**

-Does the analysis presented match the analysis plan?

-Are the results clearly and completely presented?

-Are the figures (Tables, Images) of sufficient quality for clarity?

Reviewer #1: The results are clearly in presented and the figures are of sufficient quality for clarity.

Reviewer #2: The results are clearly described and appropriate.

**Conclusions**

-Are the conclusions supported by the data presented?

-Are the limitations of analysis clearly described?

-Do the authors discuss how these data can be helpful to advance our understanding of the topic under study?

-Is public health relevance addressed?

Reviewer #1: The authors' conclusions are supported by the data and the limitations are clearly described. The public health relevance is well presented.

Reviewer #2: The conclusions are supported by the data presented, which are helpful in our understanding of the extent and the diversity of scrub typhus in Thailand. This information will be very helpful to the public health of Thailand and to countries of Southeast Asia. Limitations of the study are provided.

**Editorial and Data Presentation Modifications?**

Reviewer #1: accept

Reviewer #2: I recommend accepting the manuscript for publication with only very minor suggested changes.

**Summary and General Comments**

Reviewer #1: Extensive data related to scrub typhus epidemiology and Thailand with additional emphasis on the province with the highest incidence are presented. It is an impressive analysis of scrub typhus in Thailand, and the region of highest incidence. Clearly the evidence is that the disease is increasing in incidence. Extensive statistical analyses documented correlations with elevation, rainfall, temperature, habitats complexity, and diversity of land cover.

Reviewer #2: This manuscript is very well written and provides needed results with appropriate conclusions that readers will appreciate. 

A very few minor suggested changes, comments, and questions are provided below:

Introduction:

Line 72: Would suggest that you include the area of scrub typhus that you are talking about. That is “Scrub typhus in Asia, Australia and the Islands of the Indian and Pacific Oceans, is a potentially severe but . . .”

Line 77: Would suggest that you include with the estimated mortality rate “with a wide range (min-max) of 0–70%”. This would give the reader a better understanding that the mortality rate can be quite severe in certain locations (possibly due to variation of virulence among different strains of O. tsustsugamushi- which is not dissimilar with what has been seen with RMSF).

Line 79: Would suggest that you put “(chigger)” after the words larval stage. That is … infected larval stage (chigger) of trombiculid mites [12].

Line 462: Do you have a ref(s) for this statement (and the same statement made at the end of the abstract)? I don’t disagree with these statements, and do believe the statements should be stressed. 

General comments/concerns/questions:

Was there any way of capturing whether laborers in certain locations may have worked in large urban centers (e.g. Bangkok) but would come home for rice harvesting? 

Was there a difference in the number of rice crops per year in the different regions assessed? I would think if there were, you would guess more scrub typhus cases would be found. But if the highest incidence is in areas with only a single season, then that would suggest other behavior, ecology, epidemiology is more important.

Do you expect the number of cases to decrease with larger rice fields managed? Similarly, even in the individual rice farms that are harvested by machinery would suggest the cases of scrub typhus would decrease. If this is not the case, then again other factors then rice farming come into play.

Is land use data from 2009 appropriate for your scrub typhus incidence data collected for the subsequent 10 years?

PLOS authors have the option to publish the peer review history of their article (what does this mean?). If published, this will include your full peer review and any attached files.

Reviewer #1: No

Reviewer #2: No
---

## [Editor Report · Decision Letter 1]

18 Mar 2020

Dear Dr. Wangrangsimakul,

We are pleased to inform you that your manuscript 'The estimated burden of scrub typhus in Thailand from national surveillance data (2003-2018)' has been provisionally accepted for publication in PLOS Neglected Tropical Diseases.

Best regards,

Husain Poonawala

Guest Editor

Hélène Carabin

Deputy Editor

Thank you for submitting your revised manuscript. We are glad to inform you that we have accepted your manuscript for publication.

---

## [Editor Report · Acceptance letter]

7 Apr 2020

Dear Dr. Wangrangsimakul,

We are delighted to inform you that your manuscript, "The estimated burden of scrub typhus in Thailand from national surveillance data (2003-2018)," has been formally accepted for publication in PLOS Neglected Tropical Diseases.

Best regards,

Serap Aksoy

Editor-in-Chief

Shaden Kamhawi

Editor-in-Chief
